# Medication Logistics in Professional Homecare Organisations: An Assessment of the Practical Implementation of Regulations and Recommendations

**DOI:** 10.3390/nursrep15090332

**Published:** 2025-09-10

**Authors:** Nicole Lötscher, Christoph R. Meier, Tania Martins, Franziska Zúñiga, Carla Meyer-Massetti

**Affiliations:** 1Clinical Pharmacology and Toxicology, Department of General Internal Medicine, Inselspital–Bern University Hospital, 3012 Bern, Switzerland; 2Graduate School for Health Sciences, University of Bern, 3012 Bern, Switzerland; 3Clinical Pharmacy and Epidemiology, Department of Pharmaceutical Sciences, University of Basel, 4031 Basel, Switzerland; 4Hospital Pharmacy, University Hospital of Basel, 4031 Basel, Switzerland; 5Nursing Science, Department of Public Health, University of Basel, 4056 Basel, Switzerland; 6Institute of Primary Healthcare (BIHAM), University of Bern, 3012 Bern, Switzerland

**Keywords:** medication logistics, medication storage, homecare services, professional homecare, medication management, medication process

## Abstract

**Background/Objectives:** Patients receiving professional homecare often require support in managing their medication. In Switzerland’s legislative system, medication logistics (ordering, delivery, pickup, storage) are regulated differently by each canton, making it challenging for professional homecare organisations to comply with provisions efficiently. The present study aimed to analyse the current international literature, Switzerland’s regulations about medication logistics for professional homecare, and the current practices. **Methods:** We conducted a systematic literature review of the PubMed, Embase and CINAHL databases to identify existing international research on medication logistics by professional homecare organisations published until February 2024. The results of a structured online survey on medication logistics by professional homecare organisations in Switzerland’s German-speaking regions were compared against the medication regulations currently in place. **Results:** Ten studies were included in the review. The medication logistics processes of homecare organisations have rarely been researched, especially short-term and long-term storage. Few regulations cover medication logistics in Switzerland’s legislation, and they are often formulated non-specifically and focus on inpatient facilities. Some cantons allow centralised medication storage, others prohibit it. Only one canton explicitly permits short-term medication storage under simplified requirements. We evaluated the answers of 105 homecare organisations responding to our survey; 73.7% (73/99) of them nevertheless stored medications in the short term before bringing them to patients’ homes. Switzerland’s professional homecare organisations generally fulfil their legal requirements well. There is potential to improve the formulation of standard operating procedures for each step of the homecare medication use process, especially for cleaning medication storage sites (12/31, 38.7%) and short-term storage processes (29/56, 51.8%). **Conclusions:** There are few studies or guidelines on professional homecare organisations’ medication logistics, and they generally fail to address medication storage. Short-term medication storage is common despite most cantonal requirements being strict or prohibiting it, or not regulating it all. There is an urgent need for unambiguous, practice-oriented recommendations specific to homecare, especially for short-term medication storage.

## 1. Introduction

In Switzerland, almost 465,000 people, or 5.2% of the population, benefit from professional homecare services; many of them are over 80 years old [1,2]. According to Meyer-Massetti et al.’s review of medication-related problems (MRPs) in homecare settings, most patients aged ≥65 regularly take several different medications concomitantly. This polypharmacy—usually defined as the administration of ≥5 regularly prescribed medications—together with old age, is a common risk factor for MRPs [3]. Dilks et al. showed that 85% of frail older adults requiring healthcare services who took medication were insufficiently able to manage their medication independently; this can be due to cognitive impairments or physical limitations [4]. Professional support is often needed so they can adhere correctly to their medication. In Switzerland, professional homecare, known as Spitex in its German-speaking regions (short for hospital external care), can be requested for this purpose.

The medication management performed by homecare organisations can be categorised into up to 20 process steps [5]. Centralised medication storage at a homecare organisation’s location can be limited in time until the medications delivered to the patient’s home, also known as short-term storage. Unlimited centralised storage is known as long-term storage.

While the costs of some steps, such as medication preparation and administration, are reimbursed by health insurance, medication logistics costs, which include the ordering, delivery or pickup, and storage of medications, are not. However, as many homecare patients have limited mobility, cognitive impairment or no informal caregivers to provide support, homecare organisations often adopt a logistical role to maintain the medication supply chain.

Logistical support is further complicated by Switzerland’s heterogeneous legal framework for those services. Healthcare provision is mainly defined and legislated by the country’s 26 cantons (i.e., political regions) individually. As Switzerland is a federal state, cantonal health departments can specify regional regulations regarding the design of their health legislation and policy. Cantonal Pharmacists (each canton’s competent authority regarding therapeutic products) enforce both the federal and cantonal laws on medication [6]. Because of ambiguously worded national and cantonal legislation, Cantonal Pharmacists must interpret those regulations, which can lead to heterogeneous and partially contradictory provisions nationwide. Furthermore, the situation facing outpatient homecare organisations is not comparable with that facing the inpatient facilities (e.g., long-term care institutions or hospitals), for which the existing guidelines regulating adequate medication management were predominantly designed [7]. The Switzerland’s National Association of Cantonal Pharmacists (KAV) and cantonal associations for professional non-profit homecare organisations may only issue recommendations.

To our knowledge, the national and cantonal regulations or guiding principles for medication logistics in homecare settings in Switzerland’s German-speaking regions have never been compiled and compared, nor have they been reviewed against international practices. Furthermore, we are not aware of any peer-reviewed articles that have been published on the topic of medication logistics in the professional homecare setting.

### Aim

The objective of this study was to summarise and compare relevant national and cantonal legal provisions, recommendations, and guidelines; to evaluate current practices; and to situate these findings within a broader international context.

## 2. Materials and Methods

### 2.1. Review

We conducted a systematic literature review using a scoping review design to identify the relevant literature on professional homecare services’ medication logistics. We chose an exploratory scoping review design for providing an overview of currently published studies on the topic of medication logistics in home care. An exploratory scoping review is a methodological approach used to map the existing literature on a broad topic, identify key concepts, gaps, and types of evidence, and clarify areas for future research [8].

#### 2.1.1. Search Strategy and Eligibility Criteria

To help construct an adequate search strategy for our research question of “What are the current regulatory requirements for professional homecare organisations regarding medication logistics?” we used the four elements of the CLIP mnemonic (client group, location, information, professionals) [9]: (i) client group—the patients of professional homecare organisations living in their own homes; (ii) location—the professional homecare organisations supporting those patients’ medication management; (iii) information—the storage and handling requirements for medication used by those services; and (iv) professionals: nurses and pharmacists. As ‘location’ and ‘information’ were the most relevant elements, we developed two search blocks: Firstly, homecare services themselves, including all types of professional homecare services, and secondly, medication storage and handling. Studies were included in our review if they described one or more medication logistics process step(s) in a professional homecare setting, such as ordering, delivering or picking up medication, matching drug deliveries against drug orders, and storing medication. Publications not corresponding to these criteria were excluded. No date restrictions were applied; accordingly, this search strategy examined bibliographic databases from their inception until February 2024. The articles retrieved were screened using backward citation chasing. All the study inclusion and exclusion criteria are listed in Appendix A.

#### 2.1.2. Searching and Screening

We searched the PubMed, Embase, and CINAHL databases on 23 February 2024 using MeSH or Emtree terms and keywords prepared for our search blocks, linking those blocks with the Boolean operator “AND”. Duplicate articles were removed using Zotero^®^ 6.0 software [10]. The search strings are provided in Appendix A.

Two team members independently selected eligible articles using Rayyan^®^, a systematic review management platform [11]. A first search round screened study titles and abstracts for terms relevant to our inclusion criteria. A second search round selected relevant publications using full-text screening. Conflicts were resolved through team discussion.

### 2.2. Research into Regulations and Recommendations

To compile texts on relevant national laws regarding medication management and logistics, we searched the platform containing Switzerland’s published federal laws using relevant keywords [12]. Platforms with texts on corresponding cantonal laws from all of Switzerland’s 20 German-speaking cantons were similarly searched. The criteria for including legal texts were relevant provisions mentioning medication logistics in professional homecare organisations. If specific legislation concerning medication logistics for homecare settings were unavailable, regulations for other settings (e.g., inpatient settings) were included.

Documents available on the KAV website were also searched for relevant recommendations on medication management by homecare organisations [6].

As it is the Cantonal Pharmacists who enforce the laws on therapeutic products, they were all contacted by email, informed about our study and questioned about their standards and documentation concerning medication management by professional homecare organisations. As some Cantonal Pharmacists are responsible for several cantons, 14 were actually contacted, covering all 20 German-speaking cantons. The collected data were drawn together on a variety of charts and spreadsheets.

### 2.3. Structured Online Survey

A structured online survey in German was sent out to professional homecare organisations in Switzerland’s German-speaking regions. The survey aimed to collect data on the processes used by these targeted organisations for ordering, procuring, and storing patients’ medication, and to assess how their current practices comply with national and cantonal legal provisions and the KAV’s recommendations.

The questions were designed based on the medication process steps [5]. The questionnaire was prepared using the online survey tool Findmind^®^ [13]. Taking approximately 30 min to complete, it comprised 100 open or closed questions divided into three sections with items on demographic information, on medication logistics (with subsections on ordering, delivery, and pickup), and on the regulatory requirements and recommendations the organisations followed. Follow-up items only appeared if the appropriate response was given to the relevant lead-in item (i.e., there were no supplementary questions on storage if the organisation did not store medication centrally).

A German-language version of the developed questionnaire can be provided by the authors upon request.

The cantonal associations of professional non-profit homecare organisations were asked for their membership lists and permission to send an email containing an online link to their 306 member organisations. The questionnaire was also sent to professional for-profit homecare services via their national association’s (Association Spitex privée Suisse, or ASPS) online newsletter.

By agreeing to participate in the survey, participants gave consent, including the publication of anonymized results. An English translation of the informed consent statement is provided in Appendix A. Using the Findmind^®^ tool enabled us to analyse the anonymised answers received. We used Microsoft Excel^®^ to lay out the collected data [14].

## 3. Results

Table 1 shows an overview of all documentary sources included in this study. On an international level, 10 publications were included, while on a national level, 9 national laws and 5 documents by KAV were included. On a lower level, cantonal laws (n = 33) and recommendations by Cantonal Pharmacists (n = 5) were included.

### 3.1. Systematic Review

Figure 1 illustrates the PRISMA flow chart of our literature search and screening process (15).

The scoping review retained ten publications for inclusion in the study; nine of them were observational studies and one study was conducted as a cross-sectional study. All included studies were published between 1998 and 2022, with five studies published less than five years ago (see Table 2).

**Table 2 nursrep-15-00332-t002:** Overview of the studies retained for analysis in our scoping literature review.

Characteristics		Total ^1^	Included Studies									
			Carli Lorenzini, Sweden, 2021 [15]	Chedru et al., France, 1998 [16]	Hamre et al., Norway, 2010 [17]	Josendal and Bergmo, Norway, 2021 [18]	Kleiven et al., Norway, 2020 [19]	Lee et al., Australia, 2019 [20]	Lindblad et al., Sweden, 2017 [21]	Meyer-Massetti et al., Switzerland, 2018 [22]	Schmid et al., Germany, 2022 [23]	Tahvanainen et al., Finland, 2021 [24]
Study design	Observational study	**9**	x	x	x	x	x	x	x	x		x
	Cross-sectional study	**1**									x	
Setting	Professional homecare	**10**	x	x	x	x	x	x	x	x	x	x
Order and delivery or pickup												
Order	Unspecified order	**5**	x	x	x	x		x				
	Electronically	**2**					x					x
	By fax	**1**					x					
Ordering by…	Patient	**2**	x					x				
	Homecare organisation	**5**	x	x			x	x				x
	Community pharmacy	**2**			x	x						
Pickup by…	Patient	**1**	x									
	Homecare facility	**4**	x					x		x	x	
Delivery by…	Community pharmacy	**6**	x				x	x	x		x	x
	Blistering company	**2**			x	x						
Delivery to…	Patient	**2**						x		x		
	Homecare facility	**4**	x			x	x		x			
	Community pharmacy ^2^	**1**			x							
**Matching medication ^3^**		**2**			x	x						
Storage												
Short-term centralised storage ^4^		**1**							x			
Long-term centralised storage ^4^		**2**	x								x	
Storage at the patient’s home		**4**	x						x	x	x	

^1^ Total mentions of the respective characteristics in bold. ^2^ If medications are delivered by blistering (blister packaging) companies. ^3^ Matching medication delivered/picked up against order. ^4^ Centralised storage at homecare organisation’s facility.

**Figure 1 nursrep-15-00332-f001:**
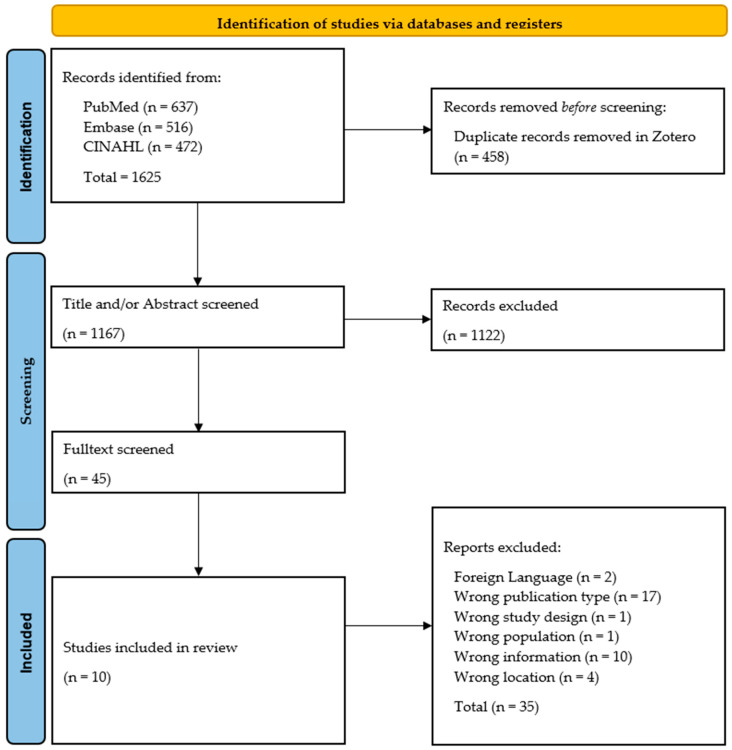
PRISMA flow chart of our systematic literature review process. Adapted with permission from PRISMA Executive. Copyright © 2024–2025, the PRISMA Executive [25].

Although ordering medication or their delivery or pickup were mentioned in every study, less than half of them addressed medication storage, focusing mainly on storage at patients’ homes (n = 4).

Some publications mentioned centralised medication storage by homecare organisations (n = 3), with one only mentioning short-term storage (see Table 1).

#### 3.1.1. Ordering Process

Seven of the studies retained for analysis described medication ordering in homecare settings, while five studies mentioned but did not define the ordering process [15,16,17,18,20], and two mentioned electronic ordering or ordering by fax (see Table 2) [19,24]. Homecare organisations ordered medication for their patients in half of the studies retained [15,16,19,20,24], but in two of them, this was only the case when patients needed support [15,20]. Medication can be packed automatically in patient-specific blister packaging, also called multidose drug dispensing. If a blister-packaging company supplies the appropriate medication, the community pharmacy responsible for the patient can order it in this form [17,18]. Josendal and Bergmo described the use of an electronic prescription tool for multidose dispensed drugs (MDD) in Norway. The E-prescribing and E-ordering of MDD led to better patient safety than conventional ordering by email or fax. After this tool’s introduction, orders were transmitted directly from the community pharmacy to the blister-packaging company, and homecare services were no longer involved in the ordering process [18].

#### 3.1.2. Delivery or Pickup

Ordered medications were mainly delivered by community pharmacies, as six studies described [15,19,20,21,23,24]. Pharmacies mainly delivered medication to the homecare organisation’s facility (n = 3). Two studies mentioned direct deliveries to homecare patients themselves [20,22], but this sometimes led to medications not yet being available when the homecare organisation made its first visit. While the complete absence of a drug is not the most frequent MRP in the overall medication use process, it is the MRP with the highest reported process impact [22]. Two studies focusing on MDD mentioned that deliveries by blister-packaging companies went either to community pharmacies [17] or directly to the homecare organisation’s facility [18].

Meyer-Massetti et al. addressed issues on this topic with regard to Switzerland’s healthcare system: in the so-called ‘self-dispensing’ cantons, physicians are allowed to distribute medications directly to their patients. This creates additional dispensing sites but can make it more difficult for homecare organisations to coordinate medication logistics [22].

Picking up ordered medication was described in less than half of the studies retained for analysis. Patients either collected their medication at their community pharmacy (n = 1) or their homecare organisations collected it for them (n = 4), mainly because homecare patients are often frail and need assistance [15,20,22,23].

#### 3.1.3. Matching Ordered Drugs Against Those Delivered or Picked up

Matching delivered or picked up drugs against what was previously ordered was hardly mentioned in the studies retained (see Table 1). The two studies only referred to deliveries of MDD, which were then checked by homecare nurses for potentially missing drugs [17,18]. Other nurses often interrupted the nurses who were checking MDD deliveries within homecare organisations as they passed by [17]. Josendal and Bergmo noted that the introduction of MDD delivery checks required four times more staff time on the delivery day [18]. The checking of medication delivered in its original packaging against the order was not mentioned in any of the included papers.

#### 3.1.4. Medication Storage

In the publications retained for analysis, medication storage in homecare settings was either not mentioned at all or only as a side note. Four publications indicated that medications were stored at patients’ homes [15,21,22,23]. Lindblad et al. briefly discussed the centralised short-term storage of MDD and non-blistered medications. They were stored in patient-specific boxes until they were brought to patients’ homes [21]. Long-term storage at a homecare organisation’s facility was mentioned once [23], and one publication suggested it by mentioning that patients’ pill boxes were filled at those facilities [15].

### 3.2. Research on Requirements and Recommendations

Switzerland’s different pieces of legislation regulate the medication management subprocesses in the homecare sector very differently. They are often formulated very vaguely, and, in some cases, they do not allow us to draw precise conclusions about the scope of their application across homecare services.

The present study retained 52 documents and legal provisions for its analyses. These included 42 national and cantonal legal provisions containing requirements relating to the steps in the medication process, regardless of how detailed the description of the subprocess was and whether homecare settings were mentioned explicitly. The KAV provided four position papers [26,27,28,29] and one guideline [7] with relevant recommendations for homecare organisations. All 14 Cantonal Pharmacists responded to our survey and provided information on how they regulated homecare medication management. Five of them provided relevant documents [30,31,32,33,34], leading to a total of 52 documentary sources.

Every step in the medication use process was regulated, at least partially, in the provisions and recommendations analysed (see Table 3). The most frequently mentioned items were external controls and inspections (n = 31), prescriptions (n = 24), and authorisation for storage (n = 16). Storage in clients’ homes was only mentioned once. Commentary on storage conditions, such as monitoring expiry dates (n = 5), temperature controls (n = 7), and hygiene requirements (n = 6) were rare, as were mentions of the ordering and delivery or pickup processes (n = 7) and matching orders against the medications delivered (n = 6).

### 3.3. National Legal Provisions and Cantonal Regulations on Medication Logistics

This section describes mandatory provisions surrounding medication management. In addition to national and cantonal laws, it also presents regulations set out by the Cantonal Pharmacists. For clarity, we only outline the requirements regarding medication logistics. The authors can be contacted for more detailed information on the legal provisions regarding other steps in the medication logistics and management process, such as the handling of prescriptions or inspections, which are of particular interest.

Both national and cantonal requirements regarding medication logistics are relevant for homecare services. Overall, four federal laws and five federal ordinances were identified. At the cantonal level, we found 33 relevant pieces of legislation in the documentation of 17 cantons. No relevant legal texts were found for three cantons. A spreadsheet overview, including all the federal and cantonal laws, ordinances, and legislation in German, can be provided by the authors upon request.

While most Cantonal Pharmacists did not have cantonal documentation regulating the medication use processes of homecare organisations, five cantons provided checklists or information sheets on correctly handling medicines in this setting. Almost all Cantonal Pharmacists orientate themselves on the KAV’s recommendations (n = 11). One Cantonal Pharmacist responsible for two cantons referred to a document prepared by another canton.

#### 3.3.1. Order, Delivery or Pickup, and Matching

As Table 1 shows, the ordering and delivery or pickup processes and the requirement to match delivered or collected drugs against the original order were only mentioned by one piece of legislation each. At the national level, it is prohibited to influence a patient’s choice from where they obtain their prescribed medications, if a healthcare provider derives an advantage from performing this, e.g., a financial gain from the sale of medicines [35]. In cantons where physicians are allowed to dispense medicines directly to their patients, they must allow their patients to collect their medication at a community pharmacy via a prescription, should they wish. One canton prohibits any attempts to influence patients on where they collect their medication, even if the influencer has nothing to gain [36].

It is explicitly stipulated by two Cantonal Pharmacists that the ordering process must be documented in writing [31,33].

There are few regulations regarding homecare patients who cannot pick up their medication at a community pharmacy. One Cantonal Pharmacist’s regulations stipulate that homecare organisations can pick up medications at the dispensing site and must bring them to their patients as fast as possible. Patients’ medication can be stored at homecare organisation facilities, but only for transit purposes. In this canton, the responsibility for that medication in short-term transit at a homecare facility lies with the original dispensing site, such as a community pharmacy, a hospital pharmacy or a physician. If dispensing sites provide delivery services, then they can be held accountable for the correct handling of the medications until the delivery to the patient’s home. Responsibilities must be written down [31].

Switzerland’s National Narcotics Act stipulates that all deliveries of narcotics must be recorded [37], and its ordinance demands records of all incoming and outgoing narcotic drugs [38].

#### 3.3.2. Medication Storage

Medication storage, in any facility, was cited several times in the legal texts retained for analysis (n = 14) (see Table 2), mainly focusing on the storage of narcotics (n = 9). These must be locked in to prevent unauthorised access [38], and cantonal regulations specify this national requirement. Some cantons limit the number of narcotics that can be stored without specifying exact quantities [39,40]. One canton permits the storage of fewer than 15 original packages of narcotics without the need for a burglary alarm system or a safe [34]. Five cantons stipulate that narcotics must be stored separately from other drugs [30,32,33,34,41]. Three cantonal laws refer to general storage, including how medications must be kept away from other goods and how access to them must be regulated [42,43,44].

#### 3.3.3. Medication Handling Supervisor

Facilities that are allowed to store pharmaceutical products centrally must employ a person who is responsible for the correct handling of medication. This medication handling supervisor is responsible for direct technical supervision and ensures that the legal regulations regarding therapeutic products are followed. The medication handling supervisor must hold a diploma in pharmacy and an official certificate proving they have no criminal record. Persons with other professional qualifications, e.g., physicians, may be considered, depending on the canton [45]. It is not clear from the wording of the national law whether homecare organisations are among the facilities that must employ a medication handling supervisor.

#### 3.3.4. Patient-Specific Storage

Storage for the sole purpose of managing medication is a term defined in several cantonal regulations (n = 8). This allows certain facilities to store medication without needing cantonal approval if it is stored individually for each patient. Although not all types of institutions that store therapeutic products are listed in all of these regulations, some mention that the provisions apply to every facility in which therapeutic products are stored [39,46,47]. Five cantons only list hospitals and/or nursing homes [41,43,48,49,50]. Only one ordinance mentions medication storage by homecare organisations specifically, stipulating that their facilities and infrastructure must be adequate to do so [51].

#### 3.3.5. Permitted and Prohibited Storage

According to the Cantonal Pharmacists, four cantons allow the central storage of medication by homecare organisations. In two of those cantons, a licence and a contract with a medication handling supervisor are required. The homecare organisations must complete a separate application form. Medications must be stored individually for each specific patient, e.g., labelled with the patient’s name, in almost all of these cantons [32,33,34]. In a third canton a consultative cooperation with a pharmacist is required. Under certain defined conditions, short-term medication storage by homecare organisations is allowed in one canton without a licence: medications must be locked in, labelled with the patient’s name and stored for a maximum of two to three days [34]. The other canton requires homecare organisations to have the same licence for short- and long-term storage. The standards applied are the same as for long-term storage in hospitals or institutional pharmacies, including employing a medication handling supervisor [30].

Five cantons’ regulations prohibit the centralised storage of medications at professional homecare organisations’ facilities because they are not specifically intended to be medicine storage facilities. One canton’s documentation provides a list of exceptions, e.g., if a patient is at risk of self-harm, medication must be stored in their home under lock and key (e.g., in a lockable drawer or box). If safety cannot be guaranteed this way, short-term storage at the homecare organisation’s facility may be necessary in exceptional cases. Storage must be performed according to the regulations: it must be time-limited, justified, and recorded. Homecare organisations are only authorised to store medications centrally on behalf of their respective dispensing sites, as the final responsibility for ensuring compliance with all the regulations lies with the dispensing site [31]. Eleven cantons have no defined standards for the short- or long-term storage of medication.

#### 3.3.6. Temperature and Expiry Dates Monitoring

No regulations on expiry dates and temperature controls were found in either national or cantonal legislation. The Medicinal Products Licensing Ordinance only mentions the need for a quality management system describing all the processes for the uniform, high-quality handling of pharmaceutical products to ensure their pharmaceutical quality [45,52]. Some Cantonal Pharmacists specifically request detailed quality management systems, including every step of the medication use process, and these must be documented [30,33,34]. This also includes storage within products’ intended temperature ranges and adherence to given expiry dates. Some cantons demand daily (2 °C to 8 °C) or weekly (15 °C to 25 °C) temperature monitoring [32,33]. Cantonal regulations rarely define expiry date monitoring, with one canton requiring this every six months [32].

#### 3.3.7. Hygiene Requirements

Only two cantonal regulations define storage room hygiene requirements [42,44], with both referring to Swiss federal food hygiene regulations, which must be applied to the handling of therapeutic products. These generally require adequate storage room ventilation and an assurance of good hygiene during and between work processes [53].

#### 3.3.8. Archiving Medication-Related Documents

There are no national legal requirements about how long documents relating to medicinal products must be kept, with the exception of a ten-year retention period for documents relating to narcotics [38]. Two cantons specify that documents regarding the handling of medication must be archived for at least five years [54,55], including files on the medication process, orders, deliveries, expiry dates, temperature monitoring, and storeroom-cleaning verifications.

### 3.4. National Recommendations on Medication Logistics

For supervising homecare organisations’ medication logistics, the Cantonal Pharmacists in Switzerland’s 20 German-speaking cantons rely mainly on documents from the KAV. The KAV provides guidelines and position statements to ensure the correct handling of medication in agreement with national laws, and it often provides additional information. We found five documents that applied to community pharmacies and inpatient facilities but were not specifically adapted to outpatient settings. Nevertheless, Cantonal Pharmacists can apply these requirements to the homecare setting. These five guidelines and position statements cover almost the whole medication use process. However, there are no recommendations for storing medicines in patients’ homes (see Table 3). A resume of the current recommendations regarding medication logistics is provided in Table 4.

#### 3.4.1. General Recommendations

Every subprocess occurring within the scope of the medication use process should be defined and documented in standard operating procedures (SOPs). The quality management system is supposed to define every step in the medication process, and each step should be documented in writing [28]. Documents relating to medications should be kept for at least one year beyond the expiry date of the respective medications, i.e., usually for at least six years [7].

#### 3.4.2. Order, Delivery or Pickup, and Matching

With the exception of the recommended SOP mentioned above, the KAV’s documents do not provide specific recommendations on ordering and delivering medication [7].

The KAV defines matching delivered or picked up drugs against the original order as a measure to ensure that only correctly delivered medications are accepted and, thus, administered to patients. This also includes, for example, checking whether transport conditions have been complied with, e.g., for goods requiring refrigerated transport. The KAV recommends documenting verification [7].

For narcotics, the date and the signature of the person performing the checks should be added to the documentation. Stock levels should be documented at the beginning and end of the year, and all receipts and issues, including the names of the patients concerned, dosage form and quantity of active ingredient, should be recorded. The medication handling supervisor should perform regular inventory monitoring [7].

#### 3.4.3. Medication Storage

Neither the KAV’s guidelines nor its position papers mention short-term medication storage. As its recommendations apply to any area in which medicinal products are stored, one can assume that its recommendations for long-term storage can also be applied to short-term storage.

According to the KAV’s recommendations, homecare organisations with centralised medication storage should hire a qualified medication handling supervisor to be responsible for this or otherwise ensure that their medicinal products are handled correctly [29]. That professional should usually be a pharmacist, but some cantons also allow a physician (with specific limitations) [27].

#### 3.4.4. Temperature Monitoring

Refrigerators and freezers for laboratory and medicinal applications should comply with DIN standard 13277:2022-05 [56]. Exceptions may be made for small quantities of stored medicines or for insulin. The KAV recommends the weekly monitoring of room temperatures and the daily monitoring of refrigerator temperatures. Temperature measuring devices should display the minimum and maximum temperature within a certain period, or have a calibrated monitoring system (electronic temperature monitoring device). The ideal location for the thermometer should be determined by temperature mapping the room or refrigerator, preferably under extreme winter and summer conditions. Thermometers should be calibrated from the beginning, and the periodic recalibration of measuring devices is necessary and should be documented. The same calibration and recalibration requirements apply to built-in measuring devices. The KAV recommends periodically checking thermometers with an alarm function. The functionality of alarm signalling systems using SMS or email should be checked monthly, at least [26].

#### 3.4.5. Hygiene and Quality Requirements

The KAV recommends that storage rooms are easy to clean to ensure compliance with hygiene regulations. Out-of-date medicines should be labelled and stored separately from other medicines until disposal [7].

### 3.5. Survey

We evaluated the responses of 105 professional homecare organisations that answered at least our mandatory question on long-term medication storage. The response rate for non-profit organisations was 33.6% (103/306). The total response rate could not be calculated, as the number of contacted for-profit private organisations is unknown. Because follow-up items only appeared if the appropriate response was given to the relevant lead-in item and because not every organisation responded fully to the survey, the total number of answers to each item varies.

The responding professional homecare organisations were from different catchment areas and of different sizes and profit status (see Table 5). The participating professional homecare organisations represented all of Switzerland’s German-speaking cantons bar one.

#### 3.5.1. Medication Use Process

Professional homecare organisations use the KAV’s recommendations (47/93, 50.5%), those of their respective cantonal associations (51/93, 54.8%), and guidelines from their Cantonal Pharmacists (47/93, 50.5%) to orient their medication management processes. Fourteen professional homecare organisations also obtained information on the handling of medications from their peers (14/93, 15.1%).

Most professional homecare organisations had written management processes (90/97, 92.8%) and employed a person responsible for quality management (90/98, 91.8%). Almost all of them had SOPs for ordering medication (84/99, 84.9%), monitoring temperatures between 15 °C and 25 °C (19/23, 82.6%), and monitoring expiry dates (26/32, 81.3%). More than two-thirds of organisations had SOPs for delivery or pickup processes (78/99, 78.8%) and long-term medication storage (27/35, 77.1%). Some subprocesses were less well defined in explicit SOPs, including monitoring storage temperatures between 2.0 °C and 8.0 °C (10/15, 66.7%), matching delivered drugs against orders (51/85, 60.7%), short-term medication storage (29/56, 51.8%), and the cleaning of storage sites (12/31, 38.7%).

#### 3.5.2. Medication-Related Documents

Almost all organisations documented their ordering process (82/96, 85.4%). There were very heterogeneous numbers of responses for documenting the monitoring of temperatures between 15 °C and 25 °C (11/16, 68.8%) and between 2 °C and 8 °C (6/8, 75.0%), monitoring expiry dates (10/32, 31.3%), matching medication deliveries against orders (44/83, 53.0%), and storage site cleaning processes (19/32, 59.4%)].

Almost half of the homecare organisations responding on these items reported that they archived their documents for five years or more, e.g., on expiry date monitoring (3/6, 50.0%), on storage room cleaning (8/18, 44.4%), on temperature monitoring (15–25 °C, 7/16, 43.6%; 2–8 °C, 3/8, 37.5%). Almost half of the organisations kept documentation on matching medication deliveries against orders for less than a year (14/32, 43.8%).

#### 3.5.3. Order

Almost every organisation carried out some ordering processes for at least some of their patients (96/99, 96.9%). The reasons mentioned were patients’ limited personal resources (e.g., mobility, cognition) and the need to ensure reliable follow-up processes.

The most common ordering methods were using emailed forms (61/96, 63.5%), via telephone (47/96, 48.9%), and via a free-text email (32/96, 33.3%). Lesser used ordering methods were ordering directly from an electronic patient system (12/96, 12.5%), using medication scanners via a USB stick on a tablet (3/96, 3.1%), ordering apps (2/96, 2.1%), sending forms by fax (2/96, 2.1%), and sending a free-text fax (1/96, 1.0%). According to the participating organisations, the main reason for using different types of ordering is that orders must be transmitted differently depending on the particular dispenser.

#### 3.5.4. Delivery or Pickup

Drugs can be dispensed and delivered by community pharmacies, mail-order pharmacies, and general practitioners (GPs). The frequency of different delivery or pickup options is shown in Figure 2. Responses from 98 participants on the question on the delivery and pickup of ordered drugs, with a total of 412 responses (multiple responses possible), were included in the graphical representation.

Drugs are mostly dispensed by community pharmacies [57]. In Switzerland, nearly every canton allows non-pharmacy entities (e.g., at emergency rooms or GPs’ practices) to dispense drugs directly to their patients [58].

Some organisations reported that certain dispensing GPs required their patients to obtain their medication exclusively from that GP’s drug stocks. This led to greater workloads for those organisations affected because they had to order drugs from other dispensing sites.

##### Delivery

Slightly more organisations stated that deliveries were made to their patients’ homes (54/98, 55.1%) than to their facility (51/98, 52.0%). Most organisations passed on the costs of delivery (19/51, 60.8%) or pickup (67/79, 84.8%) to their patients. Just over half of the professional homecare organisations surveyed had narcotics delivered to their facility (51/99, 51.5%), which required a physical signature upon receipt in 60.8% (31/51) of cases.

##### Pickup

Medication was mostly collected from community pharmacies by patients or a relative thereof (69/98, 70.4%) or by an employee of the professional homecare organisation (63/96, 64.3%). Overall, 77.6% (76/98) of professional homecare organisations stated that at least some of their patients collected their own medication from dispensing sites, whereas almost all organisations responded that they sometimes or always picked up medication for their patients (88/98, 89.8%).

#### 3.5.5. Matching Drug Deliveries Against Orders

Checking delivered or picked up drugs against orders occurred at almost every organisation’s facility (85/88, 96.6%). Some participants stated that they checked medications directly at dispensing sites where they picked them up. One organisation responded that it received almost exclusively pre-filled pill boxes, with no need to check again before giving them to patients.

#### 3.5.6. Medication Storage

Drugs can be stored at the homecare organisation’s facilities long-term or only for a limited period until they are brought to the patient’s home.

Overall, 73.7% (73/99) of the professional homecare organisations surveyed stored drugs for a limited duration. For most, short-term storage was limited to a maximum of 48 h (38/54, 70.4%), and 66.7% (70/105) of organisations did not store medications long-term. Of these organisations, two-thirds only stored drugs for short periods until they transferred them to patients’ homes (44/66, 66.7%).

Reasons for long-term storage at homecare organisations’ facilities were filling patients’ pill boxes (23/33, 70.0%), avoiding potential misuse by patients or relatives (24/33, 72.7%), and patients’ mental and cognitive impairments (29/33, 87.9%).

#### 3.5.7. Access to Stored Medication

Half of the participating organisations with short-term medication stocks short-term stored medication on open shelving or in cupboards that could not be locked (28/56, 50.0%). Most of these organisations did not monitor access to their storage rooms (19/28, 67.9%).

Access to long-term, centrally stored medication held at room temperature was more frequently regulated. Most of these organisations locked up their medications (30/34, 88.2%) using a key or a code (16/30, 53.3%), a key box (10/30, 33.3%) or a badge (4/30, 13.3%) to open storage cupboard doors. Most of the refrigerators for long-term cooled medication storage could not be locked (10/17, 58.8%), but 70.0% (7/10) of those organisations had their refrigerators in lockable rooms.

#### 3.5.8. Storage of Cooled Medication

Half of the organisations with long-term storage capacity stored refrigerated medications (17/34, 50.0%) and, of these, 55.8% (10/17) stored medications in pharmaceutical refrigerators, 29.4% (5/17) used food refrigerators where medicines were the only products stored, and 11.8% (2/17) used food refrigerators where medicines were stored alongside other products. The majority stored medication in the centre of their food refrigerators (5/7, 71.4%) or at the bottom (5/7, 71.4%). One organisation stored its medicines in the refrigerator door (1/7, 14.3%). No products were stored under the freezer compartment.

#### 3.5.9. Temperature Monitoring

Most organisations regularly monitored room temperatures (25/35, 71.4%) and refrigerator temperatures (15/17, 88.2%). Room temperatures were mostly monitored on a weekly basis (10/15, 66.7%) and, in some cases, even more frequently (4/15, 26.7%). Refrigerator temperatures were mostly checked weekly (6/8, 75.0%) or even less frequently (1/8, 12.5%). Less than half of the organisations checking the temperature regularly used electronic temperature recorders for continuous monitoring [15–25 °C (8/24, 33.3%); 2–8 °C (7/15, 46.7%)]. Most organisations not using temperature recorders documented temperature monitoring manually [15–25 °C (11/16, 68.8%); 2–8 °C (6/8, 75.0%)].

#### 3.5.10. Hygiene and Expiry Date Monitoring

Almost every organisation cleaned their storage sites regularly (32/33, 97.0%). However, the frequency differed from one week to one year: weekly (1/29, 3.4%), monthly (6/29, 20.7%), every three to six months (18/29, 62.1%), and yearly (4/29, 13.8%).

Almost every organisation with centralised long-term storage conducted expiry date monitoring (32/33, 97.0%), and all those responding to our survey reported that this was performed at least every six months (n = 28). However, some stated that they only checked expiry dates when medication was being prepared for use (e.g., using a medication adherence aid).

## 4. Discussion

The overall number of publications retained for analysis was small, as only ten suitable publications were found. Since the search strategy was quite broad and intended to comprise as many publications as possible, we conclude that the number of internationally published papers on the topic of medication logistics in homecare organisations is very limited. The studies retained dealt mostly with the provision of medication in different medication dispensing systems, focusing particularly on provision and prescription. Legal regulations or recommendations were not mentioned in any of the included studies.

The homecare organisations described in the studies retained for our review were heavily involved in the ordering, collection, and delivery of medication, as were those in our survey. The survey showed that homecare organisations often used documents provided by the KAV to guide their implementation of accurate medication management systems. This highlighted the importance of those documented recommendations and demonstrated how a guideline could be adapted to an outpatient setting to make a significant contribution to standardising medication processes within professional homecare organisations.

### 4.1. Medication-Related Documents

The KAV strongly recommends documenting all the medication management subprocesses executed [28]. Three Cantonal Pharmacists request this documentation in their authorisation forms [30,33,59], and since nearly all Cantonal Pharmacists refer to the KAV’s recommendations, it may be concluded that all the medication-use process steps should be documented. While most of the organisations surveyed reported temperature monitoring, other process steps were less widely documented. Just over half of the organisations documented storage site cleaning, and nearly 70% did not document expiry date monitoring. Documenting executed tasks is important as the KAV suggests that only documented activities can really be considered to have been performed [7].

To ensure the ability to trace activities, documents should be archived for a specified duration. While national regulations clearly define a ten-year storage period for documents concerning narcotic products [38], document archiving for other medications was rarely defined. Only two cantons stipulated a retention period of five years [54,55]. Specific durations are helpful for homecare organisations so that they can properly administer their quality management systems. Less than half of the homecare organisations surveyed stored documents regarding medication processes for five years or longer. Although we did not specifically ask about the storage of documents regarding narcotics, documents on matching drug deliveries against orders were stored for less than a year in most organisations. Longer document archiving is needed, especially for documentation matching drug deliveries against orders.

### 4.2. Order

Homecare organisations are highly involved in the delivery or pickup process. In half of the included studies, homecare organisations undertake the ordering of required medication for their patients. However, most of the studies retained for our review did not define ordering processes in detail. Two studies, from Sweden and Australia, mentioned that homecare organisations only ordered medication if patients required assistance [15,20]. This was also often the case among Switzerland’s professional homecare organisations. Indeed, Switzerland’s legislation does not specifically mention professional homecare organisations’ medication ordering processes. In most cases, Cantonal Pharmacists do not demand written specifications for ordering medication, except in two cantons [31,33].

### 4.3. Delivery or Pickup

The results of the literature research and the survey demonstrated that the practices in Switzerland were comparable to those in the international setting. In six studies, homecare organisations are directly involved in the delivery or pickup process of medication, and over a half of the participating homecare organisations stated that deliveries were made to their facility. In Switzerland, by law, patients must not be influenced as to where they choose to obtain their medication [35]. However, certain participating organisations reported that not all dispensing GPs followed this regulation. If many dispensing sites are involved in the medication logistics process, homecare organisations may have additional work. In some cantons, homecare organisations must clearly specify the delivery processes used with each dispensing site in a written contract, which leads to greater workloads for those organisations [31].

### 4.4. Matching Drug Deliveries Against Orders

The KAV recommends documenting the matching of delivered drugs against orders and formulating an SOP for this process step [7]. Although just over half of the organisations have written SOPs for the check, it is one of the least frequently documented actions in the medication use process. Most of the organisations carried out these checks, especially for narcotics. If narcotics are stored at the homecare organisation’s location, verification and written documentation are mandatory [38]. The current practices comply mostly but not completely with these national provisions, as almost half of the surveyed organisations do not sign receipts when they accept a delivery of narcotics.

### 4.5. Medication Storage

Medication storage by homecare organisations was only mentioned specifically in a few pieces of legislation, and it was only a side issue in the international publications retained for our study on homecare organisations’ medication logistics. In some studies, centralised storage could only be inferred from the fact that the patient’s pill boxes were prepared at the homecare organisation’s facility. One study, from Sweden, described short-term storage occurring at a homecare organisation’s facility on a patient-specific basis. Although that study did not mention the exact durations of short-term storage at homecare organisations, drugs were stored in their facilities until delivered to patients’ homes [21]. This corresponds to cantonal regulations [59] and the handling of short-term stored medication in homecare organisations in Switzerland. Most of the organisations stored medication for less than 48 h before they carried them to patients’ homes.

#### 4.5.1. Permitted and Prohibited Storage

We found publications from two countries, which describe or suggest long-term storage at a homecare organisation’s facility. It remains unclear whether homecare organisations in the countries, where these studies were conducted, are legally allowed to store medication at their facility or not. In Switzerland, specific legal regulations for storage at a homecare organisation’s facility are missing. As a result, Cantonal Pharmacists must apply the regulations developed for inpatient care to outpatient homecare settings. This leads to heterogeneous, inconsistent regulations across German-speaking cantons. While five cantons prohibit centralised long-term medication storage by professional homecare organisations, four others explicitly allow it. One canton even has this written down in a piece of cantonal legislation [51]. Even in these four cantons, however, the requirements are not uniform. Whereas organisations in one canton can store medication for their patients for up to two to three days, under defined conditions, organisations in another canton must apply the same approval process used for long-term storage to medications they store for just a few hours. This appears to be a disproportionate requirement if we consider that the same medications can be stored in patients’ homes without monitoring.

#### 4.5.2. Patient-Specific Storage

Patient-specific medication storage would seem to be the best match for the situation faced by Switzerland’s homecare organisations. Just one of our international studies retained for analysis described patient-specific medication storage [21], and this has been requested by almost all of the cantons that allow long-term storage [30,32,33,34]. Two of these four cantons require a contract between the homecare organisation and the medication handling supervisor, and one canton only requires this if medications are stored long-term [30,32,34].

National and cantonal regulations mainly deal with access to pharmaceutical products, which have to be stored under lock and key to prevent unauthorised access. Half of the organisations in our survey storing medications short-term did not use lockable cupboards, and, in most cases, there was no mention of regulated access to the storeroom. Most organisations locked in medications stored for the long term, except for those refrigerated. Furthermore, SOPs for short-term storage subprocesses and long-term refrigerated storage were mainly missing, and not all organisations with long-term storage performed regular temperature monitoring. Indeed, hardly any specifications could be found on the cleaning processes at storage sites. We conclude that short-term storage, refrigerated storage, and storage location cleaning processes need specific regulations and recommendations appropriate to the outpatient settings in which homecare organisations operate.

#### 4.5.3. Expiry Date Monitoring

Expiry date monitoring was not mentioned in either the studies retained in our review nor in any national or cantonal regulations. Switzerland’s recommendations and regulations only imply that expiry date monitoring should be performed regularly. Just one canton explicitly stipulated that medication expiry dates should be checked twice a year [32]. The organisations responding to our survey reported that they monitored expiry dates every six months or more frequently. However, they often stated that they carried out expiry date monitoring when they were preparing patients’ pill boxes. It is difficult, therefore, to decide whether expiry date monitoring is mainly performed consciously or carried out incidentally during medication preparation. Conducting expiry date monitoring on an irregular basis could lead to homecare professionals forgetting to do so or not documenting it properly. It is better to carry out periodic and planned expiry date monitoring, e.g., every three months, to remove expired and no longer required medication, thus contributing to medication safety for homecare patients.

### 4.6. Study Limitations

As the survey was limited to Switzerland’s German-speaking regions, its applicability to other linguistic regions may be restricted. Differences in organisational practices and legal frameworks—particular to individual regions—may also limit the transferability of the findings to international contexts. The survey only contained one mandatory question (whether the homecare organisation’s facilities were used for long-term medication storage), and thus, the number of responses to each individual item varies, which can make it difficult to interpret the results overall. Despite not every participating organisation providing demographic information, they were included in the analysis if they met the inclusion criterion of answering the mandatory question.

Unfortunately, it was impossible to distribute the survey via a validated distribution list. Nevertheless, we wanted to include for-profit and non-profit organisations. While we were able to send personal email-links for access to the survey to non-profit organisations, for-profit organisations were only reached through an impersonal electronic newsletter distributed by their umbrella organisation, the ASPS, without possible follow-up. Perhaps this is why only two for-profit organisations responded to the survey. As the number of for-profit organisations contacted was unknown, it was not possible to calculate a response rate. The inability to obtain a response rate introduces uncertainty about the representativeness of the survey sample, especially in the for-profit sector. It is possible that the participants were mainly organisations with a particular interest in medication logistics and therefore paying greater attention to this issue, leading to self-selection bias. As a result, the findings may not fully reflect the views or practices of all homecare organisations in Switzerland.

In some cantons, there is only one professional non-profit organisation. As a result, the anonymous publication and discussion of cantonal-level comparisons are not possible, as the responding organisations would be identifiable.

Each canton had many of its own laws, as well as regulations and recommendations set out by their Cantonal Pharmacist. It is possible that not every existing legal text was revealed within the scope of this project. In addition, most of the legal texts discovered were written in a rather unspecific manner, often making it difficult to decide whether they were relevant to homecare settings or not. Relevance, and thus enforcement, frequently depends on the interpretations made by Cantonal Pharmacists.

The small number of publications available, that we could include in the systematic literature review, limits the strength and generalisability of our findings. The available evidence may not capture the full range of practices and challenges in medication logistics within homecare internationally. Consequently, the conclusions drawn should be interpreted with caution, recognising that the evidence base is still emerging.

## 5. Conclusions

To the best of our knowledge, this was the first study to systematically address the logistical challenges faced by homecare organisations from both legal and clinical perspectives. As the survey and the research on legal frameworks only covered Switzerland’s German-speaking regions, generalisation of the results to a population outside of Switzerland should be avoided.

One major challenge for professional homecare organisations in Switzerland is that the legislation and recommendations regarding medication management are rarely written specifically for outpatient settings; they often relate to inpatient facilities or community pharmacies. Existing regulations on centralised medication storage in homecare organisations’ facilities not only vary greatly between cantons but also contradict each other in certain cases. A national guideline overriding these decentralised regulations, written specifically for the handling of medications in the outpatient care settings covered by professional homecare organisations, may be beneficial to all of them and would facilitate the consistent implementation of appropriate medication use processes across the country.

Specifically legalising a homecare organisation’s ability to store medication in their facilities for a limited duration would close an urgent supply gap in their patients’ care, especially as those patients are becoming increasingly old, multimorbid, and unable to organise their medication themselves. Indeed, short-term medication storage seldom lasts longer than a few hours. Thereafter, medication is stored in private households anyway, without supervision or monitoring. A pragmatic solution would be preferable to the current situation, with options that either exclude centralised storage or require expensive authorisations for each homecare organisation’s facilities.

## Figures and Tables

**Figure 2 nursrep-15-00332-f002:**
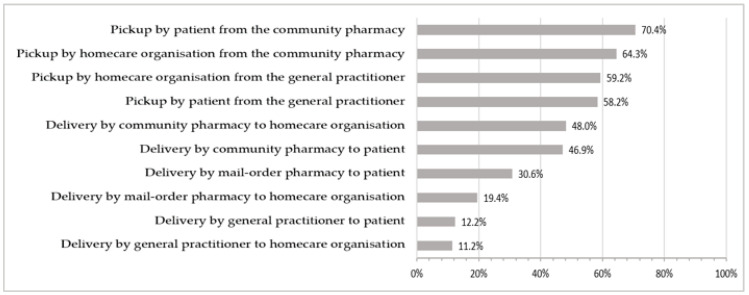
Frequency of use of the different delivery or pickup options.

**Table 1 nursrep-15-00332-t001:** Overview of international, national, and cantonal level documents reviewed.

Documentary Sources Included	Medication Logistics Process Steps Mentioned (n)		
	Medications Order/Delivery/Pickup	Matching Medications ^2^	Medication Storage ^3^
International studies (N = 10)	10/10	2/10	3/10
National laws ^1^ (N = 9)	1/9	1/9	4/9
KAV ^1^ (N = 5)	2/5	2/5	4/5
Cantonal laws ^1^ (N = 33)	1/33	1/33	10/33
CP’s recommendations ^1^ (N = 5)	3/5	2/5	4/5

N = Number of documentary sources examined mentioning medication logistics process steps. n = Number of sources mentioning medication logistics process steps. KAV = Switzerland’s National Association of Cantonal Pharmacists. CP = Cantonal Pharmacist. ^1^ Laws and documents referring to Switzerland. ^2^ Matching medications ordered against medications delivered/picked up. ^3^ Medication storage at a homecare organisation’s facility.

**Table 3 nursrep-15-00332-t003:** Overview of mentions of medication management process steps in the included sources.

	Medication Process Steps															
	Prescription	Order and Delivery/Pickup	Matching Orders Against Deliveries	Storage Permission	Storage at Patient’s Home	Centralised Storage	Centralised Storage of Narcotics	Medication Handling Supervisor	External Controls	Access Arrangements	Storage Requirements	Expiry Date/Temperature Monitoring	Hygiene Requirements	Medication Provision	Returns, Disposals	Documentation
Sources Included																
National laws ^1^	x	x	x	x			x	x	x	x	x		x			x
KAV ^1^	x	x	x	x		x	x	x	x	x	x	x	x	x	x	x
Cantonal laws ^1^	x	x	x	x		x	x	x	x	x	x		x		x	x
CP’s documents ^1^	x	x	x	x	x	x	x	x	x	x	x	x		x	x	x

KAV = Switzerland’s National Association of Cantonal Pharmacists. CP = Cantonal Pharmacist. ^1^ Laws and documents referring to Switzerland.

**Table 4 nursrep-15-00332-t004:** Recommendations on regulated processes in medication logistics.

Process Step	Recommendation
Quality control	SOPs for ordering, delivery/pickup, matching delivered/picked-up items against the order, temperature monitoring at 2–8 °C and 15–25 °C, expiry date control, storage-site cleaning Documentation of each step in the medication use process
Ordering, delivery/pickup, matching	Verification of all incoming medication upon delivery/pickup against the order/prescription Documentation of incoming and outgoing narcotics
Short-term storage	Regulated access to medications (no open shelfs) Medical refrigerators dedicated exclusively to medication storage Temperature monitoring with alarm signalling systems Daily temperature checks at 2–8 °C/weekly checks at 15–25 °C Monthly cleaning of storage sites (including refrigerator defrosting)
Long-term storage	See “short-term storage” Expiry date control every 3 months

SOP = Standard operating procedure.

**Table 5 nursrep-15-00332-t005:** Characteristics of the participating homecare organisations.

Characteristics	Number of Homecare Organisations Responding (n, %)
**Catchment areas**	N = 99
Urban	19 (19.2%)
Semi-urban	28 (28.3%)
Rural	52 (52.5%)
**Number of employees**	N = 96
≤20	11 (11.5%)
21–50	27 (28.1%)
51–100	29 (30.2%)
101–200	20 (20.8%)
≥201	9 (9.4%)
**Profit status**	N = 98
Non-profit public organisation	75 (76.5%)
Non-profit private organisation	21 (21.5%)
For-profit private organisation	2 (2.0%)

## Data Availability

The original contributions presented in this study are included in the article/Appendix A. Further inquiries can be directed to the corresponding author(s).

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
