# Peer review of "Medication Logistics in Professional Homecare Organisations: An Assessment of the Practical Implementation of Regulations and Recommendations"

_nursrep, 2025, doi:10.3390/nursrep15090332_

Round 1
Reviewer 1 Report
Comments and Suggestions for Authors
Thank you for the opportunity to review the manuscript entitled "The Practical Implementation of Regulations and Recommendations for Medication Logistics by Professional Homecare Organisations". This review article aims to summarize and compare 86 national and cantonal legal provisions, recommendations, and guidelines on medication logistics in professional homecare organisations, assess current practices, and place the findings in a broader international context. The topic is highly relevant and timely, with important implications for medication safety, quality of care, and professional nursing practice. The international contextualisation and the structured use of the CLIP mnemonic to guide the scoping review design are notable strengths.
The manuscript is generally clear and comprehensive and addresses a significant gap in understanding the regulatory framework guiding medication logistics in homecare settings. However, there are some areas that would benefit from clarification and improvement. The exclusion criteria do not necessarily need to be stated when they are simply the opposite of the inclusion criteria, and this point should be made clearer to avoid confusion. Ethical considerations could be further discussed, particularly questioning whether it would not be more appropriate to follow the ALLEA guidelines since the study does not involve direct human subjects. The bibliography could also be strengthened by including a greater proportion of recent references, as only 31% of the citations are from the last five years and 62% from the last ten years.
The inclusion of an example of the informed consent statement as a supplementary file would also increase transparency. The tables presented are helpful in understanding and summarizing the information.
In conclusion, this is an important and relevant article that addresses a significant knowledge gap, as no similar reviews have been published recently.
Congratulations on the work presented!
Reviewer 2 Report
Comments and Suggestions for Authors
This is a valuable paper on a topic that is very underexplored and needs to be exposed. With an ageing population and a reliance in many countries on homebased care, the issue of medication logisitics is important to support caregivers and care workers. Expanding the search to grey literature such as government or Pharmeceutical authorities may have given additional evidence to the argument. Countries like Australia have guidelines that support medication logistics that may not be included in peer reviewed searches. This might be something for a future review.
Author Response
Comment 1: This is a valuable paper on a topic that is very underexplored and needs to be exposed. With an ageing population and a reliance in many countries on homebased care, the issue of medication logisitics is important to support caregivers and care workers. Expanding the search to grey literature such as government or Pharmeceutical authorities may have given additional evidence to the argument. Countries like Australia have guidelines that support medication logistics that may not be included in peer reviewed searches. This might be something for a future review.
Response 1: Thank you very much for your appreciation of our work and highlighting this important point. We agree that expanding the search to include grey literature and national guidelines from other countries, such as those from Australia, would strengthen the discussion, and we will consider this direction for future work. We truly appreciate your constructive suggestion and insight.
Reviewer 3 Report
Comments and Suggestions for Authors
Dear Researchers, I congratulate you on your work. I have provided my general assessment and recommendations below.
The article is one of the few studies that systematically examines the issue of medication logistics in professional home care services, drawing on both international literature and Swiss legal regulations. The topic is timely and of high importance for health systems in the context of an aging population and increasing demand for home care. The research combines a review of legislation with field data (survey), offering a mixed-method approach. The findings provide practical recommendations for policymakers, legislative bodies, and practitioners.
-
The international comparative discussion of the findings could be further expanded. In particular, comparisons with countries that have similar health systems could enhance the generalizability of the results.
-
Although the scarcity of studies in the literature is methodologically well explained, the impact of this limitation on the interpretation of the results could be discussed more explicitly.
-
The policy recommendations section could be strengthened to include more concrete and actionable steps.
-
The survey response rates—particularly in the private sector—are low; the potential bias this might cause could be emphasized more strongly in the discussion.
-
Cantonal-level comparisons of the survey results could be presented in greater detail in tables or appendices.
Reviewer 4 Report
Comments and Suggestions for Authors
Concerns and Recommendations for Manuscript Revision:
Title: The term assessment should be included in the title, as the study focuses on reviewing current practices rather than evaluating an intervention.
Background/Problem Statement:
A brief report of similar studies conducted in the target population should be provided.
Medication logistics should be clearly defined and explained.
The current problem statement does not clearly capture the research gap.
The study objective should be stated clearly in a single sentence.
Methods:
A clear definition of the scoping review design should be provided with references.
The purpose of the structured online survey should be clearly explained in the methods section. The rationale for using this approach should be stated, as it is not reflected in the title or abstract.
The methods section is currently weak and lacks sufficient detail.
The manuscript should adhere to PRISMA (Preferred Reporting Items for Systematic Reviews and Meta-Analyses) guidelines for literature selection. A PRISMA flowchart should also be included to clearly illustrate the stages of identification, screening, eligibility assessment, and final inclusion of studies.
Authors may model their methodology after recent systematic reviews such as:
Reviews Evaluating Information Technology-Based Cardiac Rehabilitation Programs and Support: A Systematic Review
Biochemical Markers of Early Renal Dysfunction in Patients with β-thalassemia Major: A Systematic Review and Meta-analysis
Results:
Results should initially be presented in tables to allow readers to better understand the study conditions.
Discussion:
The discussion should be reported comprehensively rather than in a bullet-point or topic-heading format.
Limitations:
Since the included studies mainly relate to a specific region, this should be acknowledged in the limitations section.
Conclusion:
Avoid generalizing the results to populations outside of Switzerland.
